# Study on Different Shear Performance of Moso Bamboo in Four Test Methods

**DOI:** 10.3390/polym14132649

**Published:** 2022-06-29

**Authors:** Aiyue Huang, Qin Su, Yurong Zong, Xiaohan Chen, Huanrong Liu

**Affiliations:** 1Department of Biomaterials, International Center for Bamboo and Rattan, Beijing 100102, China; huangaiyue1124@163.com (A.H.); 15956609529@163.com (Q.S.); zyrong666@163.com (Y.Z.); 18664331398@163.com (X.C.); 2Key Laboratory of National Forestry and Grassland Administration/Beijing for Bamboo & Rattan Science and Technology, Beijing 100102, China

**Keywords:** shear performance, test methods, moso bamboo, fracture behavior, shearing characteristics

## Abstract

Bamboo is recognized as a potential and sustainable green material. The longitudinal-splitting and shear strengths of bamboo are weak but critical to its utilizations. To discuss the different shear performances of bamboo, the shear strength and behaviors of bamboo culm were investigated by four test methods: the tensile-shear, step-shear, cross-shear, and short-beam-shear methods. Then, the different shear performance and mechanisms were discussed. Results indicated that the shear strength was significantly different in the four test methods and was highest in the step-shear-test method but lowest in the tensile-shear-test method. Moreover, the typical load-displacement curves were different across the shear methods but were similar to the curves of the respective loading modes. The axially aligned fiber bundles played an important role in all the shear performances. In the tensile-shear method, specimens fractured at the interface of the bamboo-fiber bundles. However, compress-shear behaviors were a combination of compression and shear. Then, the cross-shear method, in compress-shear, was lower than that of the step-shear method because of oval-shaped bamboo culm sections of different thickness. In the short-beam shear method, the behaviors and shearing characteristics were like bending with the fiber bundle pulled out.

## 1. Introduction

Bamboo is widely used for buildings and other structural applications all over the world, not only for its fast growth, low cost and environmental friendliness, but also for its outstanding mechanical and physical properties [1,2,3,4]. Moso bamboo in China and Guadua bamboo in Colombia, for example, are capable of being used for dwellings and wide-span bridges [5]. However, due to the longitudinal arrangement of the fibers, bamboo has excellent tensile, bending, and compression properties in axial performance, except for its splitting and shearing properties. Longitudinal splitting is the dominant mode of failure for most bamboo in structural applications [6]. Shear failure, which is instantaneous and catastrophic, occurs in brittle materials. Therefore, the resistance to longitudinal splitting (cracking) is of fundamental importance in bamboo constructions.

Shear strength is determined by the anatomical structure of bamboo [7]. Bamboo culm, which consists of about 50% parenchyma cells, 40% fibers, and 10% vessels, is typically a bio-composite material [8]. The stiff fiber bundles embedded into the soft parenchyma tissues are arranged exclusively in an axial manner, which strongly contribute to the axial performances, but not the shearing properties, of bamboo. The axial shear strength of the internode of moso bamboo is just 13.5 MPa, only 9% of the tensile strength. In addition, it has been demonstrated that the density affects the axial-shear properties of bamboo [9,10]. The shear strengths of four-year-old moso bamboo with densities of 0.763 g/cm^3^, 0.798 g/cm^3^, and 0.853 g/cm^3^ are 21.7 MPa, 23.2 MPa, and 24.2 MPa, respectively [11]. Moreover, shear strength depends on the age of bamboo [12,13]. Shear strength first increases and then decreases with age, which could be related to modifications in density or the anatomical structure [13]. However, aside from age, shear strength also varies among species. For instance, the shear strength of *Bambusa pervariabilis* was 18.7 MPa, but that of *Dendrocalamopisis vario-striata* was 12.9 MPa [14].

Besides, the test methods and loading speed significantly affect shear-strength findings [15]. The shear strength of moso bamboo at a loading speed of 0.2 mm/min is 21.5 MPa, but it increases by 31% when the loading speed is increased to 2 mm/min. More elastic deformation occurred in bamboo at high loading speed, which would absorb part of the energy of external force and make bamboo exhibit higher shear strength. At the same time, the shape of the bamboo specimen affects its mechanical strength. According to previous studies, the tensile-shear strength, step-shear strength, and cross-shear strength of moso bamboo are 2.5–7 MPa [16,17,18], 15–21 MPa [10,11], and 11–15 MPa [19], respectively. Therefore, the shear strength varies under different test methods. In addition, most studies have only reported the shear strength of bamboo based on one method. Moreover, the fracture behavior and shearing characteristics of bamboo have not been studied in depth.

Therefore, this study compared the shear performance and shear mechanisms of moso bamboo under four test methods. Shear strengths and behaviors were also analyzed. Moreover, the shear-fracture characteristics were observed using a field emission-scanning electron microscope (FESEM). The different shear performances and fracture mechanisms of bamboo, based on the anisotropic and two-phase composite structure of bamboo culm, were also discussed.

## 2. Materials and Methods

### 2.1. Materials

Ten four-year-old moso bamboo (*Phyllostachys edulis*) culms were obtained from Huoshan, Anhui, China. Their average diameter at breast height was 100–130 mm, whereas the culm-wall thickness was 10–12 mm. The culms were air-dried for one year before the experiments. Samples were prepared from the internode sections of the whole bamboo wall, at a height of 1.5–2.0 m.

To avoid the effects of specimen’s variety on test results, the specimens for shearing performance comparisons in four test methods were prepared from one internode. The referenced standards, shapes, dimensions of specimens, test setups, and number of specimens in different test methods are listed in Table 1. Prior to tests, all samples were conditioned in a climate-controlled chamber at 20 °C under 65% relative humidity for three weeks. Additionally, except for the short-beam-shear method, where the load was applied in the T direction, the loading direction in the other three methods was in the L direction.

### 2.2. Methods

A tester (Instron series 5582, Norwood, MA, USA) equipped with a load cell with capacity of 10 kN was applied for tensile-shear and short-beam-shear tests, but a load cell with 100 kN capacity was used for the cross-shear test. Another tester (Shijin, series WDW-E100D, JiNan, China) equipped with a load-cell capacity of 10 kN was applied for the step-shear test. The test speed was controlled by displacement, in which the specimen failed in 60–90 s. After mechanical-shear tests, the fracture surfaces of each tested specimen were cut off. The fracture characteristics were investigated in detail using a field emission-scanning electron microscopy (XL30 FEG-SEM, FEI, Hillsboro, OR, USA) after the sputter coating of the specimens with gold in a vacuum chamber.

## 3. Results and Discussion

### 3.1. Different Shear Strengths in Four Shear Test Methods

The shear strengths of the bamboo culm were considerably different in the four test methods. As shown in Figure 1, the shear strength under the step-shear method in compression mode was the highest. In contrast, the lowest shear strength under the tensile-shear method was only 8.65 MPa, which is 48% lower than the step-shear strength. In addition, the cross-shear strength and short-beam-shear strength were between them.

Besides the structure, the different bamboo strengths are mainly affected by the test methods and loading modes. According to ASTM D906, the tensile-shear test was performed on tension mode. Specimens were prepared with two notches. The shear surface was the L–R section in the axial direction at the end of both notches (Figure 2a). The shear-stress concentration at the pre-notched ends increased with the tensile loading, causing shear-plane failure. The tensile-shear strength was the lowest in this way.

However, based on the ISO/TR 22157-2:2004(E) and GB/T 15780-1995 test methods, cross-shear and step-shear tests under compression were applied on round bamboo culms and bamboo strips, respectively. The axially arranged fiber bundles were beneficial to the first compression stage and played an important role on the compression shear mode. The circular bamboo culm was oval-shaped and with varied wall thickness. The shear failure occurred on the weakest side of the round bamboo culm (Figure 2d). Therefore, the cross-shear strength was only 14.1 MPa, 22% lower than the step-shear strength. The specimen dimensions of the step-shear method are shown in Table 1, where shear surface was in the gap between the top and bottom widths (Figure 2c), which caused tearing at the bottom of the specimen. This resulted in the highest shear strength (16.1 MPa). ASTM D 2344/2344M-2016 was referred to in the short-beam-shear test. Since three-point loading was applied to the specimen with a small span-thickness ratio, there was a compound-load mode of bending and shear (Figure 2b). Hence, the short-beam-shear strength was 15.4 MPa, only lower than the step-shear strength.

### 3.2. Fracture Behaviors of Bamboo in Four Test Methods

The typical load-displacement curves in the four shear methods are shown in Figure 3. The specimens showed different fracture behaviors under the test modes, which were dominantly related to the loading direction and mode. Bamboo is a typical gradient material, with fiber bundles and parenchyma cells that are axially arranged. Except for the short-beam-shear test, the loading direction of the other test modes were all parallel to the bamboo longitudinally. Bamboo lacks transverse-fiber bundles that could stop the longitudinal propagation of the crack in the tensile-shear, step-shear, and cross-shear tests, so it rapidly presented a load decrease once it reached the peak load. Additionally, although the load-displacement curves for the tensile-shear and compression-shear tests were similar for pure tension and compression. The shear changed the original fracture behavior, causing the curves to be different. In the tensile-shear test, the load-displacement curve continuously increased before failure but did not display increase and decrease patterns like for the pure-tension curves [20]. In the compression-shear, including step-shear, and cross-shear tests, the plastic-plateau stage was not notable, which was different from the pure-compression test [21]. The shear failure occurred as soon as the load peaked, followed by a sharp drop in the load. However, in the short-beam shear test, the loading direction was perpendicular to the fiber. The strong fiber prevented the propagation of the radial crack, which occurred in a stepwise manner. Thus, the fracture behavior of the short-beam shear test was like the pure-bending test [22].

### 3.3. Shearing Characteristics and Failure Mechanisms in Four Test Methods

The shearing characteristics and mechanisms of bamboo in the four test methods were investigated and analyzed from the macroscopic scale to the microscopic scale. The shearing in the four methods could be divided into axial and transverse. Axial shearing included the tensile-shear, step-shear, and cross-shear tests, whereas the transverse shear method was for the short-beam shear test. At the macroscopic scale, the shearing characteristics of the axial shear specimens were similar. As shown in Figure 4a, Figure 6a, and Figure 7a, the crack paths at the outer surface in the axial direction were straight but serrated at the inner surface (Figure 4b, Figure 6b and Figure 7b). These were caused by the horizontally arranged cells on the inner surface and the longitudinally arranged cells on the outer surface [23]. Additionally, the density of fibers decreased gradually in the radial direction, from the outer to the inner culm wall, which caused the shear-failure surface to change from smooth to rough, as shown in Figure 4c, Figure 6d, and Figure 7c. Furthermore, the main shearing characteristics were different under each test method. In the step-shear test, the tooth shape of the serrated crack at the inner surface was larger than that of the other two samples. Meanwhile, the bottom opening of the crack was significantly larger than that of the top. In the cross-shear test, the fracture always took place in one or two shear-failure surfaces (Figure 7a,c). Additionally, the crack deflected in the bamboo-culm wall (Figure 7a), accompanied by the buckled splitting of the bamboo-culm wall. From the top view, as in Figure 7c, twisting occurred at the failure position. As for the short-beam shear test, radial fracture occurred in the specimen in a manner similar to three-point bending. The crack was a zigzag at the bottom of the specimen (Figure 5b,e). Due to the gradient content of the fiber bundle, the crack was broader on the inner surface than on the outer surface (Figure 5e). In addition, there was no crack on the outer surface except the indention, which was different from the visible crack on the inner surface (Figure 5c,d).

At the microscopic scale, the shearing characteristics in the four test methods were also different. Except for the fiber pulled out in the short-beam-shear test (Figure 5f), the other test methods showed similar but different shearing characteristics. For the tensile-shear test, the fiber surface was smooth and only slightly peeled (Figure 4d). However, in the step-shear test, tearing of the fiber bundles and parenchyma cells at the bottom of the specimens occurred (Figure 6e,f). Moreover, the fibers and parenchymal cells collapsed in the cross-shear test (Figure 7d,e).

## 4. Conclusions

This study focused on the different shear behaviors and fracture mechanisms of bamboo in four different shear-test methods. The findings of this study will inform the good use and manufacturing process of bamboo culm. Results suggested that the test methods have a considerable effect on the shear strength and behaviors. The compound mode of compression and shear for the axial resulted in the maximum shear strength in the step-shear test, while the interface shear caused the tensile-shear strength to be the lowest. The loading direction also affected the shear behavior under the four test methods. The typical load-distance curves of the ensile-shear, short-beam shear, and compression-shear tests were similar to the respective loading-modes’ curves. However, the shear changed the original fracture behavior of the tension, bending, and compression. Additionally, the axial-shear-test methods caused typical interface-shear failure in the tensile-shear test and the overall tearing of fiber bundles in the step-shear test, while the parenchyma-cells collapsed in the cross-shear test. However, the short-beam-shear shearing characteristics were like bending with the fiber bundle pulled-out.

## Figures and Tables

**Figure 1 polymers-14-02649-f001:**
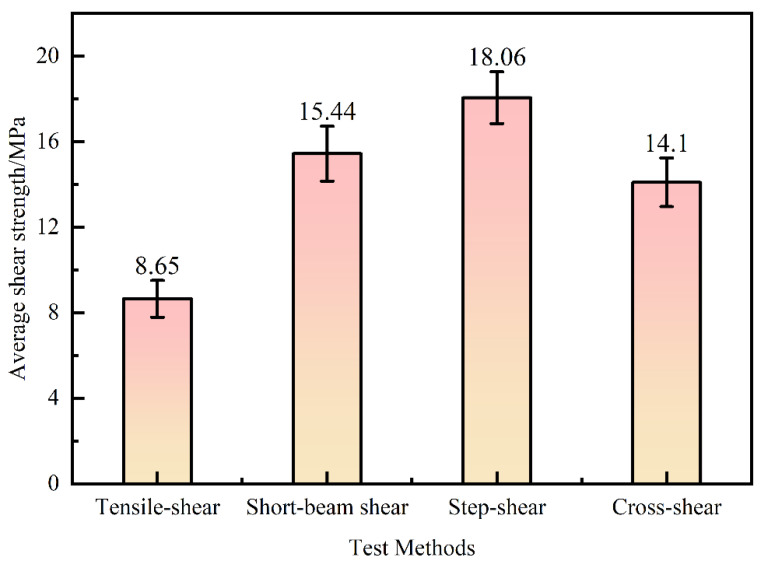
Shear strengths under different test methods.

**Figure 2 polymers-14-02649-f002:**
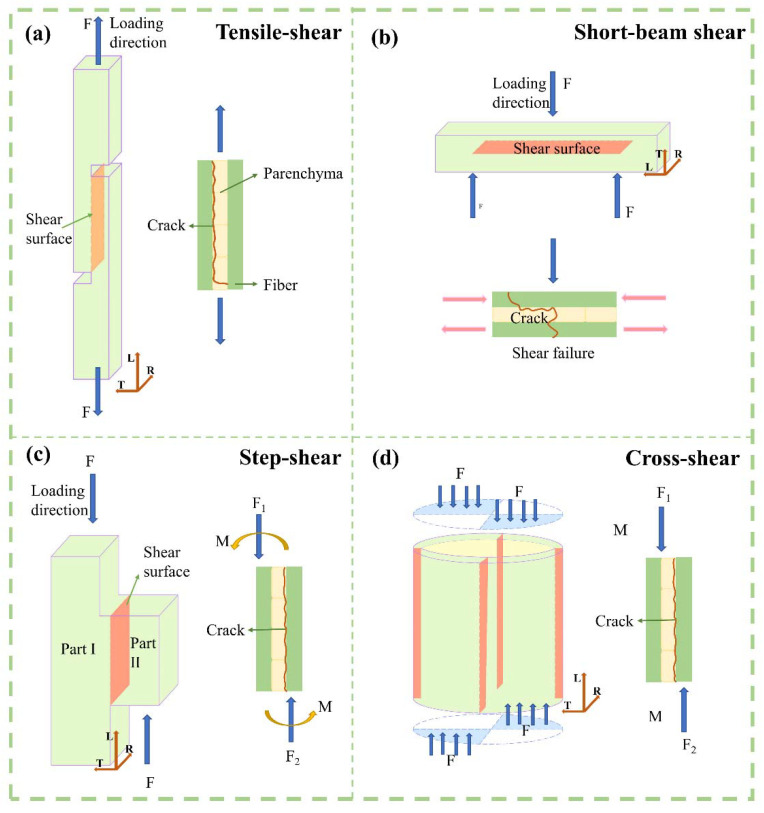
Loading modes of four test methods. (**a**) Tensile-shear test, (**b**) short-beam-shear test, (**c**) step-shear test, (**d**) cross-shear test.

**Figure 3 polymers-14-02649-f003:**
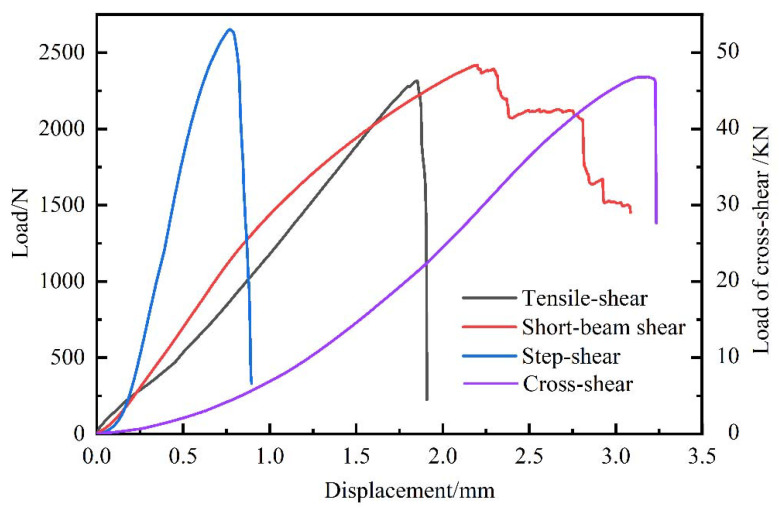
Typical load-displacement curves in four test methods.

**Figure 4 polymers-14-02649-f004:**
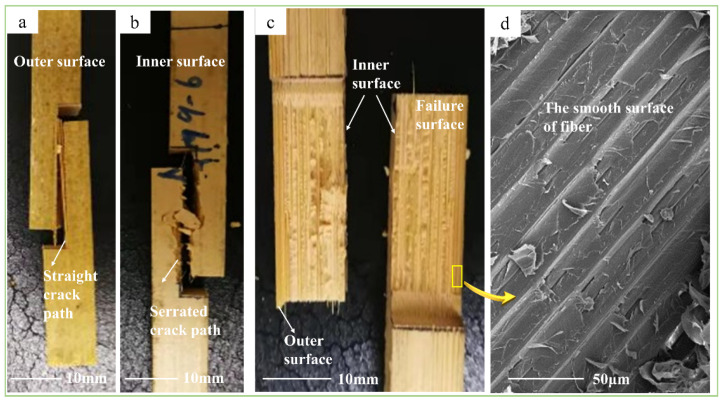
Shearing characteristics of tensile-shear method: (**a**) straight crack path at outer surface; (**b**) serrated crack path at inner surface; (**c**) shear-failure surface; (**d**) SEM image of smooth shear-failure surface.

**Figure 5 polymers-14-02649-f005:**
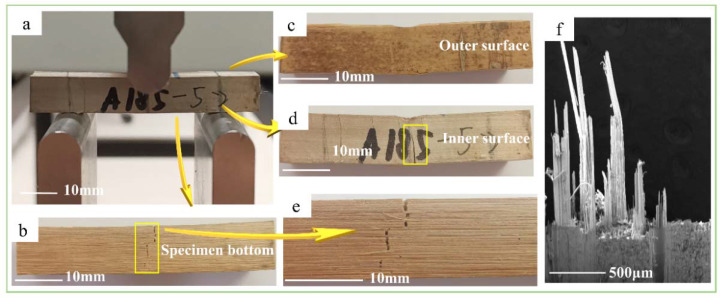
Shear-failure characteristics of short-beam-shear method: (**a**) the respective image of short-beam shear testing; (**b**) fracture in a zigzag manner at the bottom of specimen; (**c**) crack at outer surface; (**d**) straight crack path at inner surface; (**e**) a close-up look at zigzag crack path at specimen bottom; (**f**) SEM image of the pulled-out fiber bundle.

**Figure 6 polymers-14-02649-f006:**
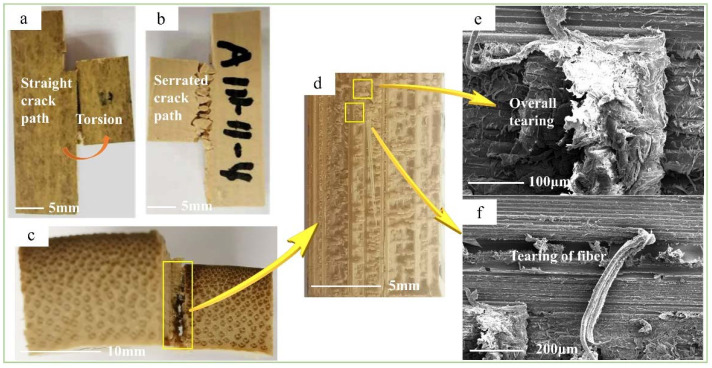
Shear-failure characteristics of step-shear method: (**a**) straight crack path at outer surface; (**b**) serrated crack path at inner surface; (**c**) Top view of failure specimen; (**d**) Shear failure surface; (**e**) SEM image of the overall tearing parenchyma cells; (**f**) SEM image of fiber tearing.

**Figure 7 polymers-14-02649-f007:**
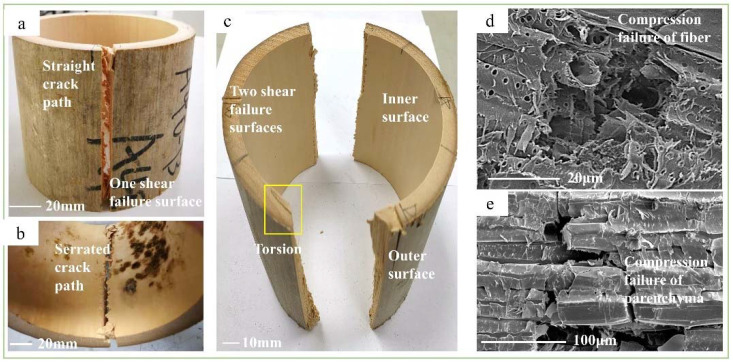
Shear-failure characteristics of cross-shear method: (**a**) failure specimen with one failure surface; (**b**) serrated crack path at inner surface; (**c**) failure specimen with two failure surfaces; (**d**) SEM image of the fiber fracture; (**e**) SEM image of the parenchyma-cell fracture.

**Table 1 polymers-14-02649-t001:** The shear-strength-test methods used in the present research.

Test Methods	Reference Standards	Quantity	Specimen Shape and Dimensions (mm)	Test Setup
Tensile-shear	ASTM D906 (Standard Test Method for Strength Properties of Adhesives in Plywood Type Construction in Shear by Tension Loading)	40		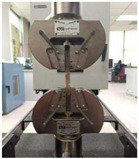
Short-beam shear	ASTM D2344/2344M-2016 (Standard Test Method for Short-Beam Strength of Polymer Matrix Composite Materials and Their Laminates)	40	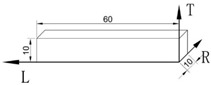	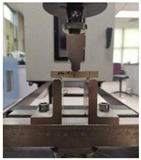
Step-shear	GB/T 15780-1995 (Testing methods for physical and mechanical properties of bamboos)	60	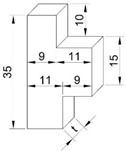	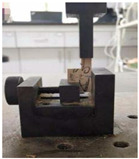
Cross-shear	ISO/TR 22157-2-2004 (Bamboo—De-termination of physical and mechanical properties—Part 2: Labora-tory manual)	10	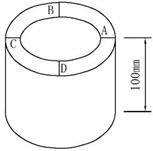	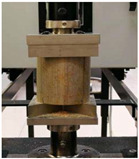

## Data Availability

Not applicable.

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
