# Peer review of "Study on Different Shear Performance of Moso Bamboo in Four Test Methods"

_polymers, 2022, doi:10.3390/polym14132649_

Round 1
Reviewer 1 Report
Nice work, but no explanation and discussion of results. What are the consequencies of the work?
Many details for testing are missing.

Reviewer 2 Report
In order to discuss the different shear performance of bamboo. In this paper, the shear strength and behaviors of bamboo culm were investigated in four test methods: tensile-shear, step-shear, cross-shear and short-beam shear modes and the different shear performance and mechanisms were discussed. The research is very interesting, the paper can be published in the journal after major revision.
(1) Scale bar is needed in all images and Error Bar is needed for the results of property.
(2) Failure mechanisms schematic diagram from different scales for the four test methods is needed to explain the difference property and failure behaviors.
Reviewer 3 Report
In this paper, the different shear behaviors and fracture mechanisms of moso bamboo were studied in four different shear-test methods: tensile-shear, step-shear, cross-shear and short-beam shear modes. The experiment design is reasonable and the research content is rich. It could be accepted for publication after the revisions.
The following comments are provided:
(1) The unit of specimens should be entered in Table 1.
(2) Table 1: Please explain whether the stress direction in step-shear test is L, R or T.
(3) Fig. 2(b): The expression of “bending failure” is not appropriate, it is suggested to change it to “shear failure”.
(4) The peak load shown in Fig. 3 appears to be inconsistent with the description in the text and Table 2, please explain.
(5) References are too simple, please add some latest references in related fields.